# Effect of a Science of Learning Course on Beliefs in Neuromyths and Neuroscience Literacy

**DOI:** 10.3390/brainsci12070811

**Published:** 2022-06-21

**Authors:** Roberto A. Ferreira, Cristina Rodríguez

**Affiliations:** Facultad de Ciencias de la Educación, Universidad Católica del Maule, Talca 3480112, Chile; mcrodriguez@ucm.cl

**Keywords:** science of learning, education, educational neuroscience, neuromyths, misconceptions, neuroscience literacy, pre-service teachers, mind, teacher training, learning styles

## Abstract

Misconceptions about the brain (neuromyths) among educators have been found across different countries, but little has been done to dispel them. The present study assessed the effect of a one-year Science of Learning (SoL) course on neuroscience literacy and beliefs in neuromyths in a sample of Chilean pre-service teachers. An experimental group of pre-service teachers, who took the SoL course as part of their university training, and a control group were needed for the study. Participants in both groups completed an online survey three times during the year (beginning, middle and end of year). The results showed that participants in both groups responded correctly to most assertions but held major misconceptions about the brain (Time 1), in line with previous studies. Regarding neuroscience literacy, participants in the experimental and control groups did not differ significantly at Time 1, but the experimental group showed significantly better performance than the control group at Time 2 and Time 3. Unlike neuroscience literacy, the results in neuromyth beliefs did not differ significantly by group at Time 1 and Time 2; however, at Time 3, the experimental group showed a significant decline in neuromyth beliefs. Overall, these results suggest that the SoL course significantly improved overall neuroscience literacy and reduced neuromyth belief among pre-service teachers, but the effect of the intervention was small.

## 1. Introduction

In the last two decades, the bridge between neuroscience and education has narrowed, thanks to the dialogue between these two disciplines, which aims to improve pedagogical practices with the use of mind and brain knowledge [1]. The dialogue between neuroscience and education has involved raising awareness of neuromyths found across different countries [2], identifying scientific knowledge that can inform education, and including cognitive psychology as a discipline that can help bridge the gap between neuroscience and education [3,4]. There is currently a lot of evidence that neuroscience knowledge can influence teachers in a positive way, for instance, increasing inquiry-based pedagogy, which leads to higher-order thinking, deep knowledge, and connections to real world problems [5,6]. Other studies have shown that teachers’ self-efficacy and student-centred practices increased after neuroscience-oriented professional development programmes [7]. Despite the evidence about the positive effect of neuroscience knowledge in education and the efforts to connect neuroscience and education through cognitive psychology, neuromyths persist [2]. Although there is still no evidence for a causal effect of beliefs in neuromyths and poor teaching practice [8,9], educational policies have already been influenced by beliefs in neuromyths, leading to large amount of resources being spent [10]. Up to date, the number of studies focused on neuroscience knowledge and belief in neuromyths are numerous, but experiments testing new ways to improve neuroscience knowledge and reduce belief in neuromyths are far fewer. Neuromyths have proved difficult to eradicate, either through a brief intervention [11] or a three-month long one [12]. The aim of the present study was to assess whether a year-long Science of Learning (SoL) course impacts general neuroscience knowledge and belief in neuromyths.

Neuroscience was first introduced into education in the USA during the 1990s, the so-called ‘decade of the brain’ [13]. Unfortunately, the initial movement caused the launch of many educational programmes claiming to be ‘brain based’ but were not actually supported by science [1]. These commercially motivated initiatives were accompanied by the emergence of misconceptions or neuromyths about the brain and its functioning, first informed by OECD in 2002 [14], and later found to be widespread across every single country that was researched [2,15,16]. These beliefs are partly explained by the lack of a common language between neuroscience and education [17]. For a decade or so, this phenomenon has been receiving a fair amount of attention as it may adversely affect teachers’ educational practices and, ultimately, education quality [18,19]. Researchers and practitioners have responded by stating the need for a more open dialogue between teachers and neuroscientists [20,21] to promote the use of reliable neuroscientific information in the classroom with the aim of improving overall teaching practices [1,22]. This is relevant because neuroscientists deal with the neurophysiological basis of behaviours such as learning, memory, attention, and motivation, which are key to education [21,23].

The new dialogue between neuroscience and education has brought criticism, such as the perception that education is better informed by psychology alone and not neuroscience [24]. This led to an interesting debate where counterarguments pointed out that mind and brain information are complementary and both can serve education, given that research in education and neuroscience involves the collaboration of neuroscientists, psychologists, and educators [23]. Several new terms have been coined to name this new discipline including ‘neuroeducation’, ‘educational neuroscience’, ‘mind, brain and education’, and more recently, the ‘Science of Learning’ [1], although the latter has been in existence long before the advent of educational neuroscience. The more dialogue between neuroscience and education has made it clearer that not only neuroscience is needed in education, but also other scientific disciplines directly connected to neuroscience. This has implications for theoretical models as well, with a progression towards including more disciplines, bridging neuroscience and education from a conventional neuroscience–education model to neuroscience–cognitive psychology–education model proposed by Bruer [3] to a more recent expanded model that includes neuroscience, cognitive psychology, educational psychology, and education [12]. Im and colleagues [12] argue that the link between neuroscience and cognitive psychology has been very successful, as indicated by the impressive number of new cognitive neuroscientists. However, the bridge between cognitive psychology and education remains elusive, indicating the requirement of another discipline (educational psychology) mediating between cognitive psychology and education [12]. They also argue that educational psychology and education share the idea of conducting research, particularly instructional interventions on realistic classroom settings. At the same time, educational psychology and cognitive neuroscience both deal with basic mechanisms of memory, learning, and share the commitment for rigorous experimental control. This path responds to the call by Bruer [4], who pointed out the need to improve connection of cognitive psychology and education, which might be the root cause of neuromyths’ persistence.

*“The neuroscience literacy of trainee teachers”* was probably one of the seminal publications on neuromyths in education. In this conference paper, Howard-Jones et al. [25] sought to understand how teachers at the beginning of their profession viewed the brain and its development. They found that trainee teachers had major misconceptions about the brain, which originated from circulating information in public spheres and brain-based educational programmes. Since then, the work by Dekker et al. [26] has had quite an impact for being the first journal paper that studied neuromyths both in the United Kingdom and the Netherlands. They confirmed what the OCDE [14] had warned a decade ago: the presence of many misconceptions about the brain among teachers in both countries. Later studies on both pre-service teachers and in-service teachers have replicated the same findings across different countries in Europe [2,15], Asia [27,28], North America [11,29], and South America [20,30]. Misconceptions are still present after almost two decades since alarms about neuromyths were raised by OECD [14]. The ones repeatedly found across countries and cultures include visual, auditory, and kinaesthetic (VAK) learning styles (e.g., Individuals learn better when they receive information in their preferred learning style), learning preferences based on hemispheric dominance (e.g., Individuals are left and or right brainers), and educational kinaesthetic or Brain Gym (e.g., Co-ordination exercises can improve integration of left and right brain functions) [31]. Another widespread neuromyth is one which advocates the importance of stimulus-rich environments in early childhood. This is based on the idea that there are critical periods for learning after which humans are not able to acquire certain knowledge [32].

Given that neuromyths are still present in education, it is crucial to know the protective factors that could prevent neuroscience and learning misconceptions to consolidate in a teachers’ belief system. Protective factors associated with the reduction of neuromyths include completing many neuroscience courses [33] or semesters of neuroscience instructions [34], reading peer-reviewed scientific journals [33], having a broader educational background [28], or general knowledge of the brain [35]. It is worth noting, however, that only having a general knowledge of the brain has also been associated with an increase belief in neuromyths [20,26,30]. The question remains as to how much knowledge of neuroscience is needed to successfully eradicate neuromyths.

Despite the need to prevent neuromyth beliefs among pre-service and in-service teachers, to date, very few studies have undertaken such a challenge [11,12,36,37,38]. In the study carried out by Im et al. [12], 50 s-grade South Korean pre-service teachers received an educational psychology course, while 49 pre-service teachers were not enrolled in the course. The course included appropriate psychology content for education (e.g., contents about cognitive, social, and emotional development; learning, memory, and complex cognition) and a section about relevant neuroscience knowledge (e.g., structure of neurons, the function of the cerebral cortex, the nature of myelination, and the controversy surrounding brain-based learning). The material used by the course lecturer included a textbook and lecture notes. In addition, a survey was administered at the beginning and at the end of the semester (three months later). The survey measured neuroscience literacy and neuromyth beliefs with items organized in six sections: General knowledge, brain function, brain development, brain structure, neuroimaging, and application of neuroscience results. Authors also collected information about the participants’ background: age, gender, number of biological sciences courses taken, and sources consulted to learn about neuroscience research. The results showed that participants in the experimental group experienced pre-post gains in neuroscience literacy across all sections except in general knowledge. However, the post-test showed that neither of the two groups experienced a significant decrease in belief in neuromyths. Additionally, participants’ background information did not modulate their neuroscience literacy or belief in neuromyths. They indicated that the neuromyths persisted after the course, suggesting that educational psychology, although necessary to bridge neuroscience and education, may not be enough. In this regard, the need for a broader instruction is recommended, including other disciplines such as cognitive psychology, developmental psychology, statistics, and experimental design. Unlike Im et al.’s study [12], Menz and colleagues [38] showed that pre-service teachers enrolled in an educational psychology course reduced psychological misconceptions, including neuromyths, after taking standard or refutation lectures with the latter being most successful. The authors concluded that refutations in psychology lectures are effective at reducing misconceptions with lasting effects. Other researchers have also demonstrated that even refutation texts are effective to target endorsement of misconceptions among in-service teachers [36]. However, the authors warned that their effectiveness is short-lived and they are not successful at preventing in-service teachers from using educational methods based on misconceptions about the brain addressed in the study. In another investigation, McMahon and colleagues [11] assessed the effect of a 90-min workshop training on general knowledge and beliefs in neuromyths of 130 pre-service English teachers. The intervention was divided into four sessions. In the first session, pre-service teachers were asked to reflect on the ‘seductive allure of neuroscience explanations’ [39]. In the second session, they were provided with information about the anatomy of key brain regions and their functions, emphasizing the connection of the two hemispheres through the corpus collosum. In the third session, neuromyths such as ‘learning styles’, ‘Brain Gym©’, and ‘left brain/right brainers’ were critically addressed. Finally, participants were given a science literacy session (fourth session) to dispel the belief that fish oils improve academic achievement and in which a brief neuroplasticity overview was provided. McMahon and colleagues [11] implemented Dekker et al.’s survey [26] with certain adaptations. The results showed no differences in mean scores before and after general knowledge intervention, whereas beliefs in neuromyths showed a slight decline (reversed scored, Mpre-test = 3.88; Mpost-test = 4.14), affecting the most prevalent neuromyths such as ‘differences in hemispheric dominance (left brain, right brain)’, ‘learning styles’, etc. Researchers also observed an increase in uncertainty when answering post-test questions in comparison with the pre-test. This result, according to the authors, was desirable as it reflects less susceptibility to false claims from participants.

The results by McMahon and colleagues [11] are promising since they showed that beliefs in neuromyths decrease in the post-test. However, this was a short intervention and there was no control group, making it hard to assess how successful the intervention was since the effect may simply be attributed to the test or the fact that the participants were aware of the intervention. There are also doubts regarding the neuroscience knowledge. If the intervention did not impact neuroscience knowledge, then is it possible for neuromyths beliefs to be affected? Taken together, the results from all intervention studies suggest that it is still not clear whether neuromyths can be eradicated after short interventions specifically targeting neuromyths [11] or after longer interventions given that the results are mixed [12,38].

The present study investigated the effect of a one-year Science of Learning (SoL) course on neuroscience knowledge and beliefs in neuromyths. The course contents covered various aspects of biosocial, cognitive, and psychosocial development as well as brain development from early childhood to adolescence. It comprised a course book and journal articles mainly from developmental psychology, cognitive psychology, educational psychology, genetics, and neuroscience fields (see Appendix A). There were two research questions: 1. Does knowledge of neuroscience increase during and after taking a SoL course? 2. Do beliefs in neuromyths decrease during and after taking a SoL course? We investigated these questions in a sample of Chilean pre-service teachers given that no previous interventions had been reported in a Latin American country and Chile offers opportunities for long interventions, given that pedagogy programmes last 4–5 years. Participants were assigned to an experimental or control group and were assessed three times throughout the year using the survey by Dekker and colleagues [26].

## 2. Materials and Methods

### 2.1. Participants

Participants included preservice teachers from two private universities in Chile. They were enrolled in a five-year pedagogy programme. The experimental group consisted of 43 (35 Female, mean age = 19.3 years, SD = 2.4) pre-service teachers taking a specially designed Science of Learning (SoL) course (see course content below). The control group included 46 (36 Female, mean age = 20.1 years, SD = 3.2) students who did not take the SoL course. Participants in both groups completed the survey at the beginning (March 2020), middle (July), and end (November) of the year. Ethical approval for this study was obtained from the Scientific Ethics Committee of Universidad Católica de la Santísima Concepción, Chile.

### 2.2. Survey

A Spanish version of the instrument created by Dekker and colleagues [26] was used (see Appendix A). This survey contains a list of 32 statements about learning and the brain, of which 17 are assertions about the brain and 15 correspond to neuromyths. The survey presents participants with three alternatives: Correct, Incorrect, or Do not know.

### 2.3. Science of Learning (SoL) Course

The course was partially based on the coursebook by Berger [40], ‘The developing person through childhood and adolescence’, and also included scientific articles to support the textbook contents. The course materials included reading chapters and papers, lecture attendance, completion of assignments, group work, group discussions, tests, and exams. Broadly speaking, the course covered biosocial, cognitive, and psychosocial development from early childhood to adolescence with a focus on brain development. It also included a section on scientific methods and how research is conducted, particularly brain research. More specific contents included synaptic plasticity, sleep and memory, language and the brain, brain maturation and development throughout the lifespan, intelligence, the senses, emotions, motivation, learning difficulties and special needs, and context variations/differences (see Appendix A, for details). The course lasted an entire academic year, split into two terms of 16 weeks each, with 2 h and 40 min of class per week including assessment.

### 2.4. Procedure

Participants completed an online survey as part of a class. The title of the survey was “Neurosciences in School” and was presented as an investigation on student teachers’ general knowledge of neuroscience. The terms neuromyth and neuroscience literacy were never used in the survey, so participants did not search for these terms online. For the pre-test, participants were first asked to read the consent form and click Yes if they agreed to take part in the study. Then they were required to fill out a form with demographic and professional background information (gender, age, etc.). Participants were not told that there would be two more successive tests throughout the year. They first completed the survey online (using Google forms) in March at the beginning of the academic year. Participants were presented again with the same survey two weeks after the last class of term one and two weeks after the last class of term two. Survey completion took around 15 min on average.

### 2.5. Data Analysis

Accuracy to items corresponding to neuroscience knowledge (1, 3, 6, 8, 10, 11, 13, 14, 16–20, 23, 29, 31, 32) and errors to neuromyth items (2, 4, 5, 7, 9, 12, 15, 21, 22, 24–28, 30) were analysed separately (see Appendix A, to view each item). All data analyses were conducted using R, version 4.0.3 [41]. Generalised linear mixed-effect models (GLMM) with cross-random effects for subjects and items [42] were used to assess the effect of time (1, 2, 3) and group (experimental, control), and the interaction between these factors on neuroscience knowledge (NK) and beliefs in neuromyths (BN). The r2beta function of the R2glmm package [43] was used to report semi partial R squared (R^2^)—a standardized measure of effect size for individual predictors. Both NK and BN were binary variables. For NK, 1 corresponded to correctly responding to each neuroscience statement and 0 represented incorrect statements. For BN, 1 represented an incorrect response to each statement, whereas 0 corresponded to correct responses or no belief in the neuromyth. Lme4 Package version 1.1–7 [44] was used for these analyses and lmerTest [45] was used to obtain *p*-values. Statistical significance and corresponding coefficient estimates (CE) were reported for accuracy (neuroscience knowledge) and errors (beliefs in neuromyths).

## 3. Results

Descriptive data showed that participants at Time 1 responded correctly to 77.5% and 79.9% of the assertions about the brain, in the control and experimental groups, respectively. However, they were unable to identify 56.7% (control group) and 61.8% (experimental group) of the neuromyths at Time 1 (see Table 1).

### 3.1. Neuroscience Literacy

The results of the generalised mixed model on accuracy (1, 0) showed no significant effect of group at Time 1 (CE = 0.14, SE = 0.20, z = 0.69, *p* > 0.05, *R*^2^ = 0.009), and no effect at Time 2 (CE = −0.09, SE = 0.18, z = −0.48, *p* > 0.05, *R*^2^ = 0.00), and Time 3 (CE = −0.02, SE = 0.19, z = −0.09, *p* > 0.05, *R*^2^ = 0.00). However, there was a significant interaction between group and Time 2 (CE = 0.51, SE = 0.25, z = 2.02, *p* < 0.05, *R*^2^ = 0.001) and Time 3 (CE = 0.62, SE = 0.25 z = 2.50, *p* < 0.01, *R*^2^ = 0.001). Furthermore, Bonferroni-corrected multiple comparisons showed no differences between the groups at Time 1 (CE = 0.10, SE = 0.19, z = 0.53, *p* > 0.05, *R*^2^ = 0.00), but a significant difference at Time 2 (CE = 0.62, SE = 0.18, z = 3.53, *p* < 0.001, *R*^2^ = 0.008), and Time 3 (CE = 0.74, SE = 0.18, z = 4.11, *p* < 0.001, *R*^2^ = 0.01). See Figure 1. 

### 3.2. Beliefs in Neuromyths

Neuromyth statements showed a lot of variation regarding beliefs. For instance, at Time 1, neuromyth 15 was endorsed by all participants in the experimental group and neuromyth 4 was believed by none of the participants in the control group. Ten of the neuromyths in both the experimental and the control groups were believed by more than 40% of the participants at Time 1 and at Time 2. However, at Time 3, this number was reduced to eight in the experimental group and remained at ten in the control group. Four neuromyths (7, 12, 15, and 21) showed above 15% belief reduction at Time 3 in the experimental group. In the control group, only one neuromyth showed over 10% reduction (25) and one (24) exhibited a 28.5% belief increase. See Table 2.

The results of the generalised mixed model on errors (1, 0) showed no overall effect of group at Time 1 (CE = 0.08, SE = 0.20, z = 0.41, *p* > 0.05, *R*^2^ = 0.00), Time 2 (CE = −0.01, SE = 0.18, z = 0.04, *p* > 0.05, *R*^2^ = 0.00), and Time 3 (CE = 0.14, SE = 0.19, z = 0.74, *p* > 0.05, *R*^2^ = 0.00). The interaction between group and Time 2 was not significant (CE = −0.10, SE = 0.25, z = −0.42, *p* > 0.05, *R*^2^ = 0.005). However, there was a significant interaction between group and Time 3 (CE = −1.02, SE = 0.25, z = −4.16, *p* < 0.001, *R*^2^ = 0.005). Bonferroni-corrected multiple comparisons between groups at each Time showed no differences between the groups at Time 1 (CE = 0.11, SE = 0.21, z = 0.52, *p* > 0.05, *R*^2^ = 0.00), no significant difference at Time 2 (CE = −0.04, SE = 0.19, z = −0.22, *p* > 0.05, *R*^2^ = 0.005), but a significant difference at Time 3 (CE = −0.81, SE = 0.20, z = −4.13, *p* < 0.001, *R*^2^ = 0.02). See Figure 2.

Given that the effect size was small, we also conducted a generalised mixed effect model on performance change in neuromyth beliefs from Time 1 to Time 3 across groups. We found a significant effect of group (CE = −17.93, SE = 3.3, *t* = −5.49, *p* < 0.001, *R*^2^ = 0.25) and a much higher effect size. Participants in the experimental group significantly reduced their beliefs from Time 1 to Time 3 in comparison to the control group. Additionally, we also conducted a Chi-squared test (χ^2^) to assess whether the proportion of participants who reduced their beliefs in neuromyths was equal between groups at Time 1, Time 2, and Time 3. The results of the Chi-squared tests showed that the groups did not differ at Time 1 (χ^2^(1) = 3.07, *p* = 0.08, Cramer’s V = 0.05) or Time 2 (χ^2^(1) = 0.11, *p* = 0.73, Cramer’s V = 0.01), but at Time 3 there was a significant association between group and accuracy (χ^2^(1) = 12.40, *p* < 0.001, Cramer’s V = 0.10). Participants in the experimental group were more likely to reduce their beliefs in neuromyths than their counterparts in the control group.

## 4. Discussion

This study investigated the effect of a one-year Science of Learning (SoL) course on neuroscience literacy and neuromyth beliefs of pre-service teachers in Chile. This is the first time an intervention was carried out in a Latin American country. Previous work had focused on the prevalence of neuromyths among pre-service and in-service teachers [20,30].

Descriptive data showed that participants at Time 1 responded correctly to 77.5% and 79.9% of the assertions about the brain, in the control and experimental groups, respectively. This is higher than the results found by Gleichgerrcht et al. [20] in a large sample of Latin American teachers (66.7%), teachers from the United Kingdom (67%), and The Netherlands (73%) [26]. It indicates that participants in our sample had substantial general neuroscience knowledge prior to the intervention, as measured by Dekker and colleagues’ survey [26]. Despite their knowledge, pre-service teachers in our study were unable to identify 56.7% (control group) and 61.8% (experimental group) of the neuromyths at Time 1. These figures are also higher than those found in a number of previous studies focused on in-service teachers [2,20,26]. Some researchers have warned that the high rates of neuromyths among teachers might be due to the questionnaires used in most studies and not because teachers adhere to neuromyths in realistic situations. Tovazzi and colleagues [46] found that using a new method based on scenarios for neuromyth detection caused the percentage of teachers who adhere to neuromyths to be much lower than when more traditional surveys are used [15,26,47]. This implies that if pre-service teachers were to teach, they would not necessarily base their teaching practice in false beliefs.

The results of the generalised mixed-effect model on neuroscience literacy found that participants in the experimental and control groups did not differ significantly at Time 1, but the experimental group showed significantly better performance than the control group at Time 2 and Time 3. Unlike neuroscience literacy, neuromyth beliefs displayed a different trajectory. The groups did not differ significantly at Time 1 or Time 2; however, at Time 3, the experimental group showed a significant decline in neuromyth beliefs. This effect was also confirmed when comparing neuromyth belief change from Time 1 to Time 3. Additionally, results from the Chi-squared test showed that participants in the experimental group were more likely to reduce their neuromyth beliefs than those in the control group. Our findings mirror those of Im and colleagues [12] since they also found that the experimental group showed an increase in neuroscience literacy after taking an educational psychology course. Here, we extend those findings by adding that the gains are incremented linearly with successive exposures to neuroscience knowledge. It also implies that improvement in neuroscience literacy could be obtained after one term, but there is further improvement after a second term, which explains why shorter interventions have not resulted in significantly more neuroscience knowledge [11].

Regarding beliefs in neuromyths, some studies have found belief reduction after taking a brief workshop [11], an Educational Psychology course including standard or refutation lectures [38], or even after being exposed to refutation texts [36]. However, other studies have found no decrease in neuromyths after an entire Educational Psychology [12] or Cognitive Neuroscience [37] course, even though gains in conceptual understanding were reported in the latter. Our findings are more aligned with those of Im and colleagues [12] and Grospietsch [37] since an effect of the SoL course was found only at the end of the year and not after the first part of the course, suggesting that neuromyth decline, by means of a Science of Learning (SoL) course, may take longer than expected.

There are a lot of similarities between the contents of the SoL course and the educational psychology course by Im and colleagues [12]. Up until Time 2, our results are also very similar (increase in neuroscience literacy but no reduction of beliefs in neuromyths). Im and colleagues attribute this surprising finding to the fact that educational psychology is just one of the disciplines that helps bridge neuroscience and education, so broader instruction in other fields such as cognitive psychology, developmental psychology, statistics, and experimental design may be necessary. A similar argument may be used to explain our results since the SoL course was intended to provide pre-service teachers with psychology and neuroscience knowledge but did not include scientific knowledge from all bridging disciplines. The fact that we found a decline in neuromyths at Time 3 for the experimental group suggests that the extra term was useful for students to abandon some misconceptions, but not all. We believe that, at least for courses that do not target neuromyths directly, course length does make a difference in neuromyth beliefs, provided that neuroscience knowledge is hard to grasp, especially for pre-service teachers that do not have science literacy training [16,18,48].

Although the participants in the current study scored high in neuroscience knowledge and those in the experimental group improved after taking the course, overall neuromyth decline was modest and 11 of the neuromyths displayed less than 10% variation across Time 1 and Time 3. The most emblematic cases were ‘Differences in hemispheric dominance (left brain, right brain) can help explain individual differences amongst learners’ (−3%), ‘Exercises that rehearse co-ordination of motor-perception skills can improve literacy skills’ (−9.2%); and ‘Short bouts of co-ordination exercises can improve integration of left and right hemispheric brain function’ (−8.3%). For instance, the neuromyth of “left and right brainers” partly stems from neuropsychology findings and neuroimaging, showing lateralization of cognitive skills such as language [33]. According to Macdonald and colleagues [33], neuromyths based on misunderstanding or over-exaggerated empirical findings are the hardest to dispel. It is possible that neuromyths of this type require learning complex information about brain anatomy, brain connectivity, and brain specialization to be eradicated. This is in line with the findings from a review of recent interventions to dispel neuromyths in Education [10]. Researchers found that exposure to a psychology [49] or a cognitive neuroscience course [37] did not produce a decline in neuromyth beliefs in undergraduate or pre-service teachers. Even after “high exposure” to neuroscience content, including taking several courses, the evidence suggests that a decrease in neuromyths is modest [33]. Grospietsch and colleagues [37] claim that neuromyths are deeply rooted in personal belief systems and taking a neuroscience course only implicitly deals with neuromyths. If participants feel confident about their deeply encoded misconceptions, eliminating them is even more difficult [50].

The SoL course as well as the interventions reported above [37,49] did not specifically target neuromyths. It is possible that neuromyths could be eradicated in a shorter period of time if they are specifically targeted. This was the case of two recent studies that used refutation lectures (more successful than standard lectures) [38] and refutation texts [36] where neuromyths were directly targeted. Both studies were successful at reducing neuromyths, but Ferrero and colleagues [36] warned that even though refutation texts were successful, they were short-lived and did not reduce the intention to use educational methods based on misconceptions about the brain, which has been attributed to a backfire effect motivated by the amplification of personal beliefs when they are confronted with counterevidence [51]. Another study that was successful at reducing the endorsement of neuromyths was that of McMahon et al. [11], which only employed a brief workshop, but apart from including neuroscience knowledge relevant to neuromyth beliefs, they also directly challenged problematic neuromyths such as visual, auditory, or kinaesthetic (VAK) learning styles, Brain Gym©, and left brain/right brain learning.

It is important to acknowledge that neuromyths and valid scientific knowledge may coexist in student teachers’ minds [10], so, even in light of the evidence, some pre-service teachers will never change their views. On the contrary, they will be likely to make use of more misinformation in line with their position to strengthen their personal convictions [51,52]. Indeed, it has been found that teachers are also more likely to make use of intuitive sources of information other than scientific knowledge to support neuromyth statements [53,54]. Another obstacle to debunking neuromyths is the fact that VAK-like approaches and the Brain Gym© method are offered to in-service teachers in schools and/or universities for teacher education [55]. Although we do not have data from Chile in this regard, it is possible that along the SoL course participants in our study were taking courses that included neuromyths or fostered neuromyth beliefs.

Among those neuromyths that decreased at least 15% after completing the SoL course were ‘There are critical periods in childhood after which certain things can no longer be learned’ (−16.8%), ‘Individuals learn better when they receive information in their preferred learning style (e.g., auditory, visual, kinaesthetic)’ (−30%), ‘Environments that are rich in stimulus improve the brains of pre-school children’ (−45%), and ‘We only use 10% of our brain’ (−25%). Despite the overall small effect size, this suggests that the SoL course was effective at debunking some neuromyths that are well-established. Previous studies have found that reading scientific journals is associated with reduced neuromyth beliefs [15,30,33]. We did not measure this variable in the present study, but the SoL course included various scientific articles to support each lesson and students were also encouraged to search for information from reliable sources. It is also possible that these neuromyths were indirectly challenged during the SoL course, which may have contributed to their decrease. For instance, current theories of language learning promote the use of ‘sensitive’ [56] rather than ‘critical periods’ [57]. The book by Berger [40] used as a textbook in the SoL course also discusses sensitive periods for language learning and states that “scientists once thought that early childhood was a critical period for language learning” [40]. This reinforces the idea that including the misinformation followed by the correction is more effective than simply presenting corrective information alone [58], even if students are not explicitly told that believing in critical periods for language learning is currently a neuromyth. Other neuromyths such as ‘We only use 10% of our brain’ are easier to debunk even if the myth is not directly challenged, given that understanding basic brain function necessarily involves learning that the brain as a whole is always active. Even though this myth is prominent worldwide, participants in the current study showed a 25% reduction in beliefs at the end of the SoL course.

Some limitations of the present study should be outlined. Given that the main difference between previous research and our study was course length, we attributed neuromyth decrease to the extended approach used. However, it is important to acknowledge that other factors such as teaching qualities of the lecturer, student gender, or student age could also be modulating the results. Our study did not offer the possibility of accurately assessing the effect of these variables, so further investigation is needed.

Course alignment and survey statements were difficult to achieve because neuroscience contents tend to be recursive, appearing more than once throughout the year. This made it difficult to track changes in neuromyth beliefs and lock them to a specific lesson within the course. Another limitation identified is the use of Dekker and colleagues’ instrument [26], which has the advantage of having been used quite extensively, allowing comparisons across studies, but it lacks a robust factorial structure [8]. Future studies could move away from classical approaches and incorporate item response theory (IRT) to more accurately assess individual items. Another issue associated with the survey is that it only measures general neuroscience literacy and participants tend to score high even before the intervention. In the present study, both groups scored very highly on neuroscience literacy at Time 1, which left little room for improvement over time. New instruments should include more challenging neuroscience statements to measure beyond general knowledge. Tovazzi and colleagues [46] also argued that new neuromyth assessment methods, not surveys, should be implemented because student teachers might report they adhere to, for instance, the ‘critical period’ hypothesis, but not necessarily transfer this belief to teaching practice. Another limitation of the current study was the low effect size, which we attribute to the sample size and the high variability across items in the survey. Hence, our results should be regarded with caution given that low statistical power can adversely affect the likelihood that our findings actually reflect a true effect [59]. Finally, the control and experimental groups were closely matched on age, gender, and programme of study, but some differences between the groups could still exist given that it was not a fully randomised controlled trial experiment.

## 5. Conclusions

The present study assessed the effect of an extended (one-year) intervention on neuroscience literacy and beliefs in neuromyths. We found that the extended intervention significantly improved overall neuroscience literacy and reduced neuromyth belief in pre-service teachers, but the effect was small provided the relatively small size and the high variability of the neuromyth statements. We are confident, however, that some of the most popular neuromyths did show a significant decline over time of at least 15%. Our findings are in line with previous evidence suggesting that several courses might be needed to successfully eradicate neuromyths when these are not confronted directly with counterevidence. To date, refutation-based interventions may be successful at debunking neuromyths in a shorter time, but their effect might be short-lived, and, in some cases, they might lead to belief reinforcement. Some neuromyths in the present study experienced a decline over time because they were indirectly challenged during the intervention. Neuromyths that did not experience a change over time were only dealt with at an implicit level, and since these misconceptions are deeply rooted in belief systems, they were not affected. Future studies should combine extended interventions and refutation activities (lectures, workshops, seminars) in which neuromyths are directly confronted with neuroscience evidence. Measurements should be taken several times during the interventions in order to assess possible backfire effects. Finally, we also recommend including activities that directly challenge neuromyths in psychology, neuroscience, or science of learning courses aimed at pre-service teachers. Ideally, these activities should be spaced over time and repeated in order to allow enough time for consolidation.

## Figures and Tables

**Figure 1 brainsci-12-00811-f001:**
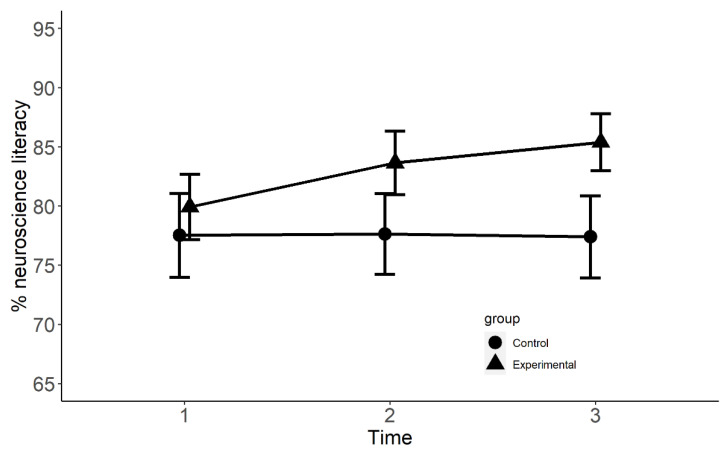
Neuroscience literacy of control and experimental groups at Times 1, 2, and 3.

**Figure 2 brainsci-12-00811-f002:**
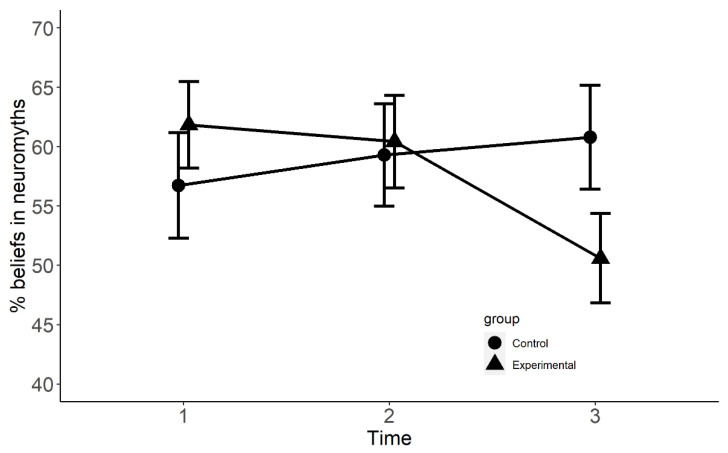
Percent beliefs in neuromyths of control and experimental groups at Times 1, 2, and 3.

**Table 1 brainsci-12-00811-t001:** Neuroscience literacy and beliefs in neuromyths of control and experimental groups at Times 1, 2, and 3.

Neuroscience Knowledge			
Group	Time 1	Time 2	Time 3
Experimental	79.9%	83.7%	85.4%
*Incorrect responses*	13.6%	14.2%	14.5%
Control	77.5%	77.6%	77.4%
*Incorrect responses*	13.2%	13.2%	13.2%
Beliefs in neuromyths			
Experimental	61.8%	60.4%	50.6%
*Correct responses*	9.3%	9.0%	9.1%
Control	56.7%	59.3%	60.8%
*Correct responses*	8.6%	8.9%	7.6%

Note: The total number of correct responses for neuroscience knowledge was 17. The total number of possible errors for neuromyths was 15.

**Table 2 brainsci-12-00811-t002:** Percent beliefs in each neuromyth in the experimental and control groups at Times 1, 2, and 3.

	Experimental				Control			
Neuromyths	T1	T2	T3	T1-T3 Change	T1	T2	T3	T1-T3 Change
2. Children must acquire their native language before a second language is learned. If they do not do so neither language will be fully acquired	33	44	36	1.4	13	15	12	−1.1
4. If pupils do not drink sufficient amounts of water (D6–8 glasses a day) their brains shrink	12	17	9	−3	0	12	14	13.8
5. It has been scientifically proven that fatty acid supplements (omega-3 and omega-6) have a positive effect on academic achievement	96	97	89	−6.6	96	100	97	0.6
7. We only use 10% of our brain	48	43	24	−25.2	56	57	66	10.2
9. Differences in hemispheric dominance (left brain, right brain) can help explain individual differences amongst learners	96	98	93	−3.1	97	97	97	−0.2
12. There are critical periods in childhood after which certain things can no longer be learned	41	30	24	−16.8	77	79	77	0.2
15. Individuals learn better when they receive information in their preferred learning style (e.g., auditory, visual, kinesthetic)	100	100	70	−30.6	86	87	86	0
21. Environments that are rich in stimulus improve the brains of pre-school children	92	95	46	−45.5	97	100	100	2.6
22. Children are less attentive after consuming sugary drinks and/or snacks	78	74	77	−1.3	68	70	75	7
24. Regular drinking of caffeinated drinks reduces alertness	81	79	88	7.2	54	63	82	28.5
25. Exercises that rehearse co-ordination of motor-perception skills can improve literacy skills	97	98	88	−9.2	100	88	88	−12.5
26. Extended rehearsal of some mental processes can change the shape and structure of some parts of the brain	18	9	12	−6	16	9	11	−5.4
27. Individual learners show preferences for the mode in which they receive information (e.g., visual, auditory, kinesthetic)	3	0	12	9.1	4	5	3	−1.9
28. Learning problems associated with developmental differences in brain function cannot be remediated by education	18	13	16	−1.8	19	14	23	4.6
30. Short bouts of co-ordination exercises can improve integration of left and right hemispheric brain function	98	100	90	−8.3	100	97	94	−5.7

## Data Availability

All data are available online at Appendix A—Dropbox: https://www.dropbox.com/sh/nyh7en5h1j9i15e/AABMrfBkFQZzIIJ7idAuZ6kaa?dl=0 (accessed on 20 April 2022).

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
