# Peer review of "Effect of a Science of Learning Course on Beliefs in Neuromyths and Neuroscience Literacy"

_brainsci, 2022, doi:10.3390/brainsci12070811_

Round 1

Reviewer 1 Report

The manuscript described the evaluation of a training aimed at dispelling neuromyths through a one-year course. Although the introduction and the other texts are clear, the data analysis is really poor. Authors are encouraged to address all the points and to add analyses to their data, so that some discussions are based on evidence, rather than on intuition.

In the abstract, the description of the "gap" (that this paper is going to address) is needed: why is your work needed?

Some information is missing: How long was the SoL course? How many hours per week, how many weeks, how much time separated the third data collection from the end of the course?
I suspect that the third evaluation was made right at the end of the course.. Then, how can author talk about "learning", if the evaluation at that time was so close to the end of the training? Could it be that participants have answered positively because of what they remembered about the course (and not because they have interiorized true facts)?

line 248: "Both NK and BN were categorical".. Then how could averages be computed (in Table 1)? 

Moreover, in table 1 the labels "MRC" and "MER" need to be clarified: I think that all the "second" and fourth" lines are the errors associated to the average in lines 1 and 3.

Lines 264-275, and table 2: a statistics could be computed in order to validate the "ocular" evaluation made by the Authors. For example, chi^2 or trend (given that time is on the ordinal scale), and you could compare the two levels of Groups. Given that much of the discussion deals with these results, a statistical analysis is deserved.

In addition, the GLMM analysis suffers from a problem of power: the effect sizes are really low. This must be accounted by the limited number of participants, which is also noted as a limitation.

In the discussion, the effect of a training is just attributed to its length. I am personally not convinced at all about this attribution: by reviewing the large literature about training efficacy, there are many other parameters that may have affected the efficacy of the training, rather than its length. For example, the immediacy (how were the judgements expressed by the students about the instructor?) and the style of the instructor (frontal teaching, dialogic, laboratorial, etc...?), the interest in the subject from participants (how were the students selected? did they change their interest from the beginning to the end?), etc. 

Line 398: "this suggests that the SoL course was effective, at least for some students". Could Authors use a cluster analysis to separate the students with high and low effect? And, then, attributing this effect to some of the variables that they have studied? The paper would increase the amount of results and, in turn, increase its reliability.

The list of limitations must be moved to the "Discussion", while in the "Conclusions" some statements about the relevance of the current work and the implications of the findings are indicated.

Minor issues:

Line 140: the two sentences need to be separated (an "and" or a subject need to be added).

Lines 239 and 240 are the same: delete one..

Author Response

Dear Reviewer 1,

We appreciate the comments to our manuscript. Our point-by-point response is attached below.

Reviewer 2 Report

The authors are to be congratulated on a well-designed study that both replicates earlier studies and advances scholarly understanding by introducing an extended intervention. As the authors make clear, previous studies in the area of neuromyths and their prevention have been limited, largely dependent on short courses. So the innovation of a year-long programme is important in terms of advancing our understanding of the mechanisms underlying the defence against silly ideas.

The manuscript as a whole is very well-written, clearly organised, and the narrative of the article develops in a sensible way. My only small concern is with the conclusion. It seems to me that the findings from the study will be widely discussed. Consequently, the rather brief and superficial closing discussion does not really provide adequate justice to the richness of the paper. If the consideration of limitations is removed from the concluding section, very little remains!

Author Response

Dear Reviewer 2,

We really appreciate your comments to our manuscript. Please, find our response attached.

Round 2

Reviewer 1 Report

The Authors responded positively to many issues. Still some remain open, and deserve further elaborations. When referring to an issue already opened in the previous round, I used the same numbers as in the rebuttal letter.

7. I appreciate the inclusion of the percentage of variation from T1 and T3 included in the table. However, when Authors stated "We did not compute any statistics on individual neuromyths statements because statistics on questionnaires are usually conducted on the entire set of items", I would suggest them to check this presentation: https://statmath.wu.ac.at/people/trusch/IMPS2017/RIRT-Workshop-IMPS-2017Pt2.pdf, considering the Item Response Theory as a reliable tool to check their hypothesis.

9. let's consider a course made by a bad instructor: I think that Authors would agree that neither after Time 2 nor after Time 10, he or she would obtain any change. Thus, Authors must not pass the message that Time is "the only" relevant factor, but considering that other factors (neither manipulated, nor controlled) were also implicitly involved. For example: Andersen, J. F. (1979). Teacher immediacy as a predictor of teaching effectiveness. Annals of the International Communication Association, 3(1), 543-559.
At least, some factors related to the instructor(s) must be acknowledged in the limitations, as missing aspects that deserve further investigation.

10. A Cluster analysis requires n observations and p>=1 variables. Therefore, Authors can perform it with just one variable (that is, those data used to create figure 2, at Time 3). I understand that their goal was to test the effects of a training, but it would be intriguing if they could also tell as why. Indeed, some factors were identified by the Authors: length, content.. why not investigating also the other aspects that they have assessed (age, gender, performance at Time 1,...)? Or to evaluate, retrospectively, if there are participants that showed an effect also at time 2?

Small issues:

5. Now the label "error" is less unclear than before: implicitly, I asked to add the explanation of "MRC" and "MER" in the caption.

Line 354: "It also implies that improvement in neuroscience literacy can be 354 obtained after one term,". Change "can" into "could": the results in this manuscript tells that there is no difference at Time 2.

Line 466: Tovazzi was not the only Author of her study.. use "Tovazzi and colleagues" instead

line 478: "was the first to assess".. How can Authors be sure to be "the first to"? The risk to be disconfirmed is high, as Chris Chambers highlighted. Simply state: "The present study assessed the effect of an extended.."

Author Response

We thank Reviewer 1 for the new comments on our manuscript. Attached are our responses.
